# Measurement of Retrobulbar Blood Flow and Vascular Reactivity—Relevance for Ocular and Cardiovascular Diseases

**DOI:** 10.3390/diagnostics13233514

**Published:** 2023-11-23

**Authors:** Elsa Wilma Böhm, Nils F. Grauhan, Norbert Pfeiffer, Adrian Gericke

**Affiliations:** 1Department of Ophthalmology, University Medical Center, Johannes Gutenberg University Mainz, Langenbeckstrasse 1, 55131 Mainz, Germany; norbert.pfeiffer@unimedizin-mainz.de (N.P.); adrian.gericke@unimedizin-mainz.de (A.G.); 2Department of Neuroradiology, University Medical Center, Johannes Gutenberg University Mainz, Langenbeckstrasse 1, 55131 Mainz, Germany; nils.grauhan@unimedizin-mainz.de

**Keywords:** measurement, perfusion, retrobulbar, vasculature

## Abstract

Abnormal retrobulbar hemodynamics have been linked to the development of various ocular diseases, including glaucoma, age-related macular degeneration, and diabetic retinopathy. Additionally, altered retrobulbar blood flow has been observed in patients with severe cardiovascular diseases, including carotid artery occlusion, stroke, heart failure, and acute coronary syndrome. Due to the complex and intricate anatomy of retrobulbar blood vessels and their location behind the eyeball, measurement of retrobulbar blood flow and vascular reactivity, as well as the interpretation of the findings, are challenging. Various methods, such as color Doppler imaging, computed tomography angiography or magnetic resonance imaging, have been employed to assess retrobulbar blood flow velocities in vivo. Color Doppler imaging represents a fast and non-invasive method to measure retrobulbar blood flow velocities in vivo. While no information about vessel diameter can be gained performing this method, computed tomography angiography and magnetic resonance imaging provide information about vessel diameter and detailed information on the anatomical course. Additionally, ex vivo studies, such as myography, utilizing genetically modified animal models may provide high optical resolution for functional vascular investigations in these small vessels. To our best knowledge, this is the first review, presenting a detailed overview of methods aiming to evaluate retrobulbar blood flow and vascular reactivity in both humans and laboratory animals. Furthermore, we will summarize the disturbances observed in retrobulbar blood flow in retinal, optic nerve, and cardiovascular diseases.

## 1. Introduction

Abnormalities in retrobulbar hemodynamics have been documented in several sight-threatening ocular disorders, including glaucoma, non-arteritic anterior ischemic optic neuropathy, age-related macular degeneration, and diabetic retinopathy [1,2,3,4,5,6,7,8]. Notably, in glaucoma patients with altered blood flow in retrobulbar vessels, changes in optic nerve head blood flow were also observed [9]. Color Doppler ultrasound (CDU) was used to analyze changes in retrobulbar blood flow in patients with pathological myopia. In this patient group, markedly reduced peak systolic blood flow velocity (PSV), end-diastolic blood flow velocity (EDV), and higher resistance index (RI) of the ophthalmic artery, central retinal artery and posterior ciliary arteries, were observed compared to patients with normal vision or low and moderate myopia [10]. Moreover, it has been proposed that an increased RI of the ophthalmic artery, central retinal artery, and posterior cerebral artery may be the first sign of developing diabetic retinopathy [11]. Given the reported link between altered retrobulbar hemodynamics and retinal and optic nerve diseases, but also between cardiovascular diseases, we aimed to present the various measurement techniques of retrobulbar blood flow and vascular reactivity and to discuss their scientific, diagnostic, and therapeutic relevance.

### 1.1. Anatomy and Regulation of the Retrobulbar Vasculature

The retrobulbar region, positioned behind the eyeball, encompasses a network of blood vessels providing nourishment to various ocular structures, including the optic nerve, retina, choroid, and surrounding tissues. A major vessel of the orbit is the ophthalmic artery (OA), which serves as the initial intracranial vessel originating from the internal carotid artery (ICA). Within its short intracranial course, the OA traverses the optic canal before entering the orbit [12]. It gives rise to multiple branches, including the central retinal artery (CRA), which constitutes the first branch of the OA in 77.5% of people and which supplies the central core of the optic nerve and the inner retina [12,13]. The choroid and outer retinal layers receive their blood supply from the long and short posterior ciliary arteries (LPCA/SPCA), which also arise from the OA. The SPCA provide nourishment to the proximal choroid and to the optic nerve head, while the LPCA continue towards the distal choroid, iris, and ciliary body. These arteries form anastomoses behind the lamina cribrosa, creating the circle of Zinn–Haller [12]. Anastomoses with the CRA through the circle of Zinn–Haller are possible, and additionally, the cilioretinal artery may contribute to the blood supply of inner retinal layers [12]. Further branches of the OA include the lacrimal artery, ethmoidal arteries, the supraorbital artery, the dorsal nasal artery, the palpebral artery, and the supratrochlear artery, supplying orbital and extracranial tissues [12]. Figure 1 provides a scheme of the retrobulbar vascular anatomy. Notably, the OA exhibits a robust anastomotic network. In cases of occlusion, vision is typically preserved in 90% of individuals. However, occlusion of the central retinal artery can lead to significant vision loss [12,14].

### 1.2. Regulation of Retrobulbar Blood Flow

Retrobulbar blood flow is under the control of parasympathetic and sympathetic innervation. Parasympathetic nerve fibers originating from the pterygopalatine ganglion exert their influence on vascular tone through the release of acetylcholine, vasoactive intestinal polypeptide (VIP), and neuronal nitric oxide synthase (nNOS) [13]. Sympathetic nerve fibers, on the other hand, encompass the OA and innervate ocular structures via posterior ciliary nerves [15]. These fibers have been identified in the peripheral muscularis of the OA and its branches originating from the superior cervical ganglion [13].

Autoregulation, a physiological mechanism, ensures a constant blood flow to an organ or tissue despite changes in blood pressure. While retinal vessels lack autonomic innervation, local autoregulatory mechanisms come into play in the regulation of retinal blood flow [16]. In contrast, choroidal blood flow is predominantly regulated by the autonomic nervous system, although autoregulatory mechanisms have also been described [17,18,19,20,21]. In the retrobulbar region, autoregulation plays a vital role in maintaining adequate blood flow to the optic nerve, retina, and other tissues while safeguarding them from damage caused by fluctuations in blood pressure. Furthermore, compensation for changes in ocular activity is possible [22]. There are two distinct types of autoregulatory mechanisms: static and dynamic. Static autoregulation encompasses neurogenic, myogenic, and metabolic factors. The myogenic mechanism relies on the ability of vascular smooth muscle cells to constrict or dilate in response to changes in blood pressure, thereby maintaining constant vascular resistance and blood flow. The metabolic mechanism is based on the tissue’s ability to regulate blood flow by releasing vasodilator or vasoconstrictor substances in response to changes in oxygen and nutrient demands. The neurogenic mechanism relies on the nerves’ ability to regulate blood flow by releasing neurotransmitters that control the tone of vascular smooth muscle cells [22]. Dynamic autoregulation acts swiftly to intercept sudden variations in perfusion pressure [22,23]. It has been shown that a decrease in systemic blood pressure induces peripheral vasoconstriction in the OA [24], indicating sympathetic vasoconstriction. After a step decrease in systemic blood pressure, decreased flow velocities were observed in the CRA and posterior ciliary arteries. This phenomenon was more prominent in the CRA than in the posterior ciliary arteries, suggesting better autoregulatory properties in the vascular bed distal to the CRA [25]. During maximal physical exercise, ocular perfusion pressure and blood flow velocities increase, while blood flow velocities in the CRA and SPCA remain stable, implying the presence of autoregulatory mechanisms in these retrobulbar vessels [26]. Numerous studies examining retrobulbar blood flow in healthy individuals have indicated the involvement of autoregulatory processes in maintaining ocular blood flow under physiological conditions, as described below.

In this review, we present methods for measurement of retrobulbar blood flow in humans and experimental animal models, while also describing changes in retrobulbar blood flow associated with ocular and cardiovascular diseases. This study should help clinical scientists as well as experimental investigators to identify the advantages and limitations of individual measurement methods and to define suitable methods for designing their research projects in the field of retrobulbar vascular tone and blood flow regulation. To the best of our knowledge, this is the first comprehensive review summarizing contemporary state of the art concerning retrobulbar blood flow measurements under physiologic and pathologic conditions.

The literature was identified via search on PubMed. The following keywords were used: (“retrobulbar blood flow” OR “ophthalmic artery” OR “central retinal artery” OR “posterior ciliary artery”) AND (“measurement” OR “color Doppler imaging” OR “computed tomography angiography” OR “magnetic resonance imaging” OR “myography” OR “transmitted light microscopy”). The research was performed from 11 March 2023 to 30 June 2023 with the following inclusion criteria: all studies, written in English, and published after 1990. The reference list of the selected articles was reviewed for further identification of relevant studies.

## 2. Assessment of Retrobulbar Blood Flow

### 2.1. Color Doppler Imaging

#### 2.1.1. Technical Aspects

Color Doppler imaging (CDI) is a non-invasive and commercially available method used to evaluate retrobulbar blood flow. It combines B-scan ultrasonography with Doppler technology, providing both anatomical and velocity information. By analyzing the time between the emission and reflection of ultrahigh-frequency sound waves, CDI generates anatomical information. When these sound waves encounter a moving object, their frequency spectrum is shifted, a phenomenon known as the Doppler shift. This shift in frequency is proportional to blood flow velocity, allowing for the visualization of blood flow and velocity information [27,28,29]. To assess retrobulbar vessels using CDI, the optic nerve is used as a landmark, and vessel visualization is achieved through color Doppler. Once the vessel is identified, the sample volume should be placed at the center of the vessel, with the Doppler angle positioned parallel to the vessel to record Doppler waveforms [27]. In the middle of the optic nerve, the CRA and central retinal vein (CRV) can be identified, producing a double waveform. It should be noted that separate measurement of these vessels is not possible due to their proximity. Additionally, measurements should be performed posterior to the lamina cribrosa to avoid contribution from retinal and choroidal blood flow [27]. CDI also allows for the measurement of the nasal and temporal SPCA, located on the respective sides of the optic nerve. However, individual SPCAs cannot be differentiated by CDI, and a uniform arterial pulse representing a bundle of small vessels is obtained [27]. Deeper in the retrobulbar tissue, the OA can be detected. Measurements should be taken on the temporal side of the optic nerve after its crossing [27]. A sketch of the location and Doppler waveforms of retrobulbar vessels is shown in Figure 2. CDI provides several parameters of ocular blood flow, including peak systolic velocity (PSV), end-diastolic velocity (EDV), and mean flow velocity (MFV). These parameters are independent on the Doppler angle, as changes affect both PSV and EDV concurrently [27]. The resistive index (RI) can also be calculated using the formula RI = (PSV − EDV)/PSV [27,30].

CDI offers a fast and reproducible method for evaluating blood flow velocity in larger retrobulbar blood vessels [31,32]. It is not affected by unclear ocular media, such as cataract or corneal diseases [27]. Training can further improve reproducibility [27]. However, precise knowledge of retrobulbar vessel anatomy and individual waveform characteristics is essential for accurate results, since anatomic variations can complicate measurements [27,33]. Measurement of blood pressure and intraocular pressure (IOP) before the examination is recommended, as both can influence retrobulbar blood flow velocity. It is important to avoid secondary elevation of IOP by pressing with the probe on the eyeball [27,29]. Additionally, information about medications, both for IOP lowering and systemic drugs, as well as systemic or local vascular diseases, should be gained prior to CDI. For instance, carotid artery stenosis greater than 70% is associated with reduced retrobulbar blood flow velocities [34,35]. The angle of incidence is a common source of error in CDI and should be kept below 60°. Angles above 60° can lead to significant errors in velocity calculations, and determining the angle in SPCA can be particularly challenging due to the vessel’s small size and tortuous course [27,29,36]. The reliability of RI as a measure of vascular resistance remains uncertain [37] and can be influenced by various factors, including vascular compliance, vascular resistance, and blood pressure pulsatility [27,38]. Another limitation of CDI is its inability to provide volumetric blood flow data. Accurate calculation of blood flow requires the determination of vascular diameter [39]. However, CDI does not provide quantitative data on vessel diameter. Some approaches have been attempted to estimate vascular diameter for the measurement of volumetric blood flow in the OA, but these methods have shown limited reproducibility [40,41,42].

#### 2.1.2. CDI under Physiological Conditions

Several studies have examined the physiological autoregulatory mechanisms in retrobulbar vessels using CDI. During iso-oxic hypercapnia, an increase in PSV and EDV has been observed, along with a decrease in RI in the SPCA, indicating the presence of metabolic autoregulation in retrobulbar vessels [43]. Other studies have reported increased PSV and EDV in the CRA and OA during hypercapnia, but without changes in RI [44]. Hosking et al. also found increased PSV in the CRA of healthy subjects during hypercapnia. During hyperoxia, reduced PSV and EDV in the OA of healthy subjects have been detected [45]. Furthermore, an elevation of RI has been observed during inhalation of pure oxygen [46]. Evans et al. found a reduced EDV and increased RI in the CRA during 100% oxygen breathing [47]. Moreover, pressure autoregulatory mechanisms have been investigated by CDI. During the Valsalva maneuver, increases in IOP, systolic and diastolic blood pressure and ocular perfusion pressure have been observed. PSV of the OA and SPCAs significantly decreases, while EDV increases, leading to a decrease in RI. Isometric handgrip causes decreased IOP and increased systolic and diastolic blood pressure as well as increased ocular perfusion pressure. In the OA, a decreased PSV, increased EDV, and decreased RI have been found [48]. Acute elevation of IOP above 40 mmHg in rabbits has resulted in a reduction in PSV and EDV of the OA and an increase in RI, suggesting limited autoregulatory capacity of the OA in rabbits under high-IOP conditions [49]. Maximum dynamic exercise has been associated with higher ocular perfusion pressure and increased PSV in the OA, while PSV remained stable in the CRA and SPCA, likely due to autoregulatory mechanisms in the retinal and choroidal circulations [26]. Postural changes have also influenced retrobulbar blood flow velocities, with stable flow velocities observed in the SPCA of healthy subjects, while flow velocities in the CRA and OA have been affected by changes in hydrostatic pressure [50]. Neurovascular coupling, which regulates retinal perfusion, is supposed to have an impact on retrobulbar blood flow. For example, flicker stimulation led to increased PSV and EDV in the SPCA of healthy subjects, while the CRA showed an increase in EDV but no changes in PSV [51]. However, performing flicker stimulation studies using CDI is limited by the need for closed eyelids during examination and the difficulty in controlling the amount of light stimulus [27].

#### 2.1.3. Clinical and Scientific Relevance of CDI for Ocular and Cardiovascular Diseases

Impairment of autoregulatory mechanisms in retrobulbar vessels may occur in several ocular diseases, such as glaucoma. Using CDI, no increase in blood flow velocities was observed after flicker stimulation in patients with glaucoma, unlike in healthy subjects [51]. Furthermore, patients with advanced glaucoma exhibited different Doppler parameters compared to those with preperimetric glaucoma [52]. Lower EDV and higher RI were found in the SPCA of patients with advanced glaucoma, while the CRA and OA showed lower PSV and higher RI. The interpretation of these Doppler parameters is challenging, as decreased EDV and increased RI in retrobulbar arteries in glaucoma could be a consequence of increased IOP, vasospasms, or atherosclerosis [52]. A meta-analysis by Meng et al. involving 2000 eyes with primary open-angle glaucoma (POAG) revealed similar results, suggesting that CDI could be a potential diagnostic tool for evaluating vascular features in the pathogenesis of glaucoma [53]. Intriguingly, lower retrobulbar blood flow values were detected in African descent patients compared to European descent patients with POAG, indicative of different disease processes in these patients [54]. A different meta-analysis, which compared individuals diagnosed with normal tension glaucoma (NTG) to control groups, revealed consistent findings regarding retrobulbar hemodynamics. The analysis demonstrated a decrease in PSV and EDV, accompanied by an increase in RI in the OA, CRA, and SPCAs among NTG patients. These findings suggest the potential role of ischemic retrobulbar hemodynamics in the development and progression of NTG [30]. Moreover, impairment of retrobulbar blood flow was more pronounced in patients with NTG compared to POAG [55]. Several more recent studies reported similar findings [56,57,58,59]. Combination of CDI with fluorescein angiography has demonstrated a correlation between arteriovenous passage time and the EDV and RI of the CRA. A significant decrease in both parameters was detected in patients with NTG [60]. Moreover, optic nerve head filling defects of patients with NTG were higher compared to controls [61]. Similar findings have been found in patients with POAG, indicating that combination of CDI with different techniques might help to better understand ocular circulatory changes in glaucoma [62]. Impaired retrobulbar blood flow has also been observed in pseudoexfoliation glaucoma and angle-closure glaucoma [63]. Retrobulbar blood flow changes in SPCA were similar in POAG and pseudoexfoliation subtypes [64]. The effects of IOP-lowering drugs on retrobulbar hemodynamics have yielded contradictory results. Topical treatment with beta blockers, such as timolol 0.5% and carteolol 2%, showed a significant reduction in RI of the SPCA, whereas RI in the carotid artery (CA), OA, and CRA remained stable [65]. However, other studies have found an increase in RI values of the SPCAs after topical application of timolol and a significant decrease in RI in the CRA and SPCA after the application of betaxolol, indicating varying vasodilatory effects of different beta blockers [66]. Treatment with dorzolamid 2%, a carbonic anhydrase inhibitor, improved ocular perfusion pressure and lead to an increase in EDV in major OAs [67]. No retrobulbar hemodynamic changes were found after therapy with latanoprost, brimonidine, or latanoprost and timolol or brimonidine and timolol fixed combinations [68,69]. But there are also studies that showed a significant reduction of RI in the OA and CRA after treatment with travoprost and latanoprost [70].

Similar to patients with POAG, patients with optic disc drusen (ODD) also had lower systolic and diastolic flow velocities in the CRA, OA and SPCA compared to healthy patients. Moreover, the changes in blood flow velocity in CRA correlated with the extent of visual field loss [71].

CDI has also been used to analyze retrobulbar blood flow in other ophthalmological diseases. In central or branch retinal vein occlusion (CRVO/BRVO) reduced velocities in the CRA and CRV were found. Additionally, CDI findings could assess the risk of ischemic conversion [72,73,74]. Patients with non-arteritic anterior ischemic optic neuropathy (NAION) also display reduced blood flow of the OA and internal carotid artery [75]. Moreover, reduced flow velocities of the OA were related to higher intima-media thickness (IMT) of the common carotid and internal carotid arteries on the side with AION [76]. Changes in retrobulbar blood flow parameters with lower EDV and higher RI and PI of SPCA and CRA have also been detected in patients with central serous chorioretinopathy (CSC), supporting the vasospasm hypothesis in its pathogenesis [77,78]. These changes were also detected in the unaffected eye, identifying CSC as a bilateral disease with possible systemic origin [79].

In patients with diabetic retinopathy, PSV and EDV in the CRA, OA, and SPCAs were significantly decreased compared to controls, while the RI in these eyes was significantly higher [80]. In prediabetic patients increased RI of retrobulbar vessels was detected compared to healthy controls, thereby representing the first step in the development of microangiopathy in diabetic patients [11]. Other studies have also reported reduced retrobulbar blood flow in patients with diabetes, suggesting that changes in retrobulbar hemodynamics may play a central role in the pathogenesis of diabetic retinopathy. The RI of the CRA might serve as a reliable biomarker for the severity of diabetic retinopathy [81,82,83,84,85]. CDI may help to detect patients with a higher risk to develop diabetic retinopathy [83]. Administration of calcium dobesilate (CaD) is associated with protective effects on blood retinal barrier, as well as with antioxidant and anti-inflammatory effects. CDI studies revealed improved retrobulbar blood flow parameters after treatment of patients suffering from diabetic retinopathy with CaD [86]. These findings indicate that CDI might represent an attractive tool for detecting and tracking therapeutic effects on retrobulbar blood flow during the treatment of retinal diseases. Combination of CDI with scanning laser Doppler flowmetry (SLDF) of the retina has revealed a positive correlation between the retinal measurement by SLDF and retrobulbar vessels by CDI in diabetic patients without diabetic retinopathy [87].

Pre-term neonates are at risk of developing retinopathy of prematurity. By conducting CDI, elevated flow velocities were detected in these patients and a correlation with the disease stage was found [88]. Other authors reported similar results with correlation of CDI parameters and the status of blood vessels in the ocular fundus [89,90]. Due to a larger axial length, myopic eyes are often affected by morphological and hemodynamic vascular abnormalities. These deviations frequently occur in the CRA and its intraretinal branches. Slower flow velocities of the CRA in the retrobulbar segment have been detected by CDI [91,92]. In eyes with myopic choroidal neovascularization (CNV), a significantly higher RI of the SPCA was detected, while no differences were reported in the CRA and CRV [93]. Other authors found similar results with decreased PSV and EDV in the CRA and SPCA in myopic eyes. Moreover, retinal microvascular density and choroidal vascularity detected by optical coherence tomography angiography (OCTA) were simultaneously lower, and higher severity of myopia was associated with impaired choroidal blood flow [94]. Reduced retrobulbar blood flow parameters were also related to thinning of the retinal nerve fiber layer (RNFL), retinal ganglion cell (RGC) and inner plexiform layer (GCIPL) in myopic eyes, indicative of a vascular pathogenesis of these myopia-related changes [95]. However, it remains unclear whether impaired blood flow in myopic eyes is a cause or consequence of the thinning of ocular tissues [96].

Thyroid eye disease (TED) is associated with retrobulbar inflammation, orbital fibrosis, and with enlargement of extraocular muscles and retrobulbar fat tissue [97]. Blood flow velocity in the superior ophthalmic vein (SOV) and RI in the OA has been detected to be higher in patients with TED compared to healthy controls. SOV maximum velocity has been found to represent a reliable indicator for disease activity [98]. Comparison of CDI parameters with the clinical activity score (CAS) has revealed similar results [99]. After orbital decompression surgery, the RI of the CRA and OA decreased [100]. CDI may thereby represent a diagnostic technique making it possible to survey disease activity and therapeutic effects.

CDI studies have also provided insights into impaired retrobulbar hemodynamics in patients with cardiovascular diseases. For example, patients with chronic heart failure exhibited lower diastolic velocities and higher RI in the OA compared to the control group. These findings may indicate increased orbital vasoconstriction in response to low cardiac output in patients with chronic heart failure [101]. In pregnant women, CDI-detected hemodynamic changes in the OA have been established as a reliable predictor for severe preeclampsia [102,103,104]. Patients with carotid occlusive disease demonstrated decreased flow velocities in the OA and CRA [105]. Moreover, the detection of reversed OA flow direction by CDI has shown high specificity for severe ipsilateral ICA stenosis or occlusion [106]. Reduced OA flow parameters in patients with severe carotid artery disease significantly improved after carotid stenting [107,108]. Similarly, chronic ocular ischemic syndrome resulting from severe carotid artery stenosis improved after carotid artery stenting, with significant increases in antegrade OA blood flow and PSV observed within 24 h after the intervention [109] and during long-term follow-up [110]. Changes in retrobulbar blood flow related to obesity during childhood have also been detected by CDI. PSV and EDV of the OA, CRA and SPCA were significantly lower in obese children compared to controls while OCTA imaging did not show retinal microvascular impairment [111]. Lower PSV and EDV values in the CRA and OA in adult patients with obesity were also detected by CDI. These findings also correlated with the severity of obesity [112]. Based on these findings, obesity appears to be a risk factor for ocular pathologies due to disturbed retrobulbar blood flow, and CDI may help to detect early macrovascular changes and to identify cardiovascular risk factors in this context.

Apart from cardiovascular diseases, changes in retrobulbar blood flow have also been detected in other systemic diseases that may affect the eye. For example, in patients with systemic lupus erythematosus (SLE) significantly lower blood flow velocities and increased RI in the CRA and SPCA have been detected by CDI. Changes in the bigger OA were associated with coexistent nephropathy and central nervous system vasculitis [113]. CDI studies in patients with rheumatoid arthritis (RA) have revealed similar results [114] and differentiation between active and remission RA patients was possible [115]. Reduction of retrobulbar blood flow values has also been detected in patients with Behcet’s disease, and these findings were more pronounced in patients with ocular involvement [116,117]. CDI may enable detection of possible ocular involvement before initial clinical manifestation [118]. In patients suffering from Takayasu arteritis, changes in RI in the OA, CRA and SPCA have also been detected by CDI and have been associated with a longer onset of the disease. In cases of retinal involvement, abnormal PSV in the OA has been found [119]. In patients with COVID-19 infection, changes in retrobulbar blood flow parameters have been detected compared to age- and sex-matched healthy controls. Notably, PSV was significantly lower in the OA of patients with COVID-19 infection [120]. Due to the fact that COVID-19 infection is related to ocular vascular diseases, such as microvascular derangement or retinal vein occlusions [121], these findings indicate a possible predisposition of patients with COVID-19 to ocular vascular dysfunction. Moreover, two weeks after application of an inactivated SARS-CoV-2 vaccine, a significant reduction of RI and PI in the OA, CRA and temporal-nasal PCA was detected by CDI [122]. Ocular adverse effects after receiving a COVID-19 vaccine, such as vascular thrombosis or acute macular retinopathy, have been described in several cases [123] and may be related to changes in retrobulbar vascular functions after vaccination. However, further examinations are necessary in this field.

To date, only a limited number of studies has been published on the measurement of retrobulbar blood flow using CDI in laboratory animals. In rabbits, for instance, there have been studies investigating the effects of heating the ocular surface on retrobulbar blood flow. These studies have demonstrated increased blood flow velocities in the CRA and the LPCAs following heating [124]. After induction of high IOP in rabbits, reduced blood flow velocities and higher RI were found by CDI [125]. Others found similar results with reduced autoregulatory capacity of the OA when IOP was raised above 40 mmHg [49]. The effect of sildenafil on retrobulbar hemodynamics in rabbits was also analyzed by CDI, but no changes in RI were detected in this study [126]. After topical application of timolol in rabbits, a dose-dependent increase in the RI of the OA was detected [127]. Studies in mice or rats have not been reported so far.

In conclusion, CDI is a non-invasive and rapid method for evaluating retrobulbar blood flow. In several clinical studies, detection of retrobulbar vascular hemodynamic changes by CDI was possible before clinical manifestations occurred. Therefore, CDI may represent a promising screening tool in several ocular and systemic diseases. However, diagnostic standards have not been established so far and long-term studies to determine sensitivity, specificity and repeatability of CDI are mandatory. Moreover, CDI is an attractive technique for analyzing vascular involvement in the pathogenesis of different ocular diseases and monitoring therapeutic effects on retrobulbar blood flow. However, it has limitations in reliably detecting volumetric data and changes in vessel diameter. Moreover, reproducibility can be ameliorated by training.

### 2.2. Computed Tomography Angiography

#### 2.2.1. Technical Aspects

Computed tomography angiography (CTA) is a diagnostic technique where intravenous injection of iodinated contrast media allows for precise visualization of blood vessels and body tissues. CT scans utilize X-rays and advanced computer processing to generate detailed cross-sectional images of the body. However, it is important to note that CTA carries certain risks in terms of patient safety. Adverse events associated with CTA include anaphylactic reactions, vasovagal symptoms, nausea, or disturbances in systemic blood pressure. Additionally, the impact on thyroid function and the potential risk of contrast-induced kidney disease must be considered [128]. Another aspect to consider is the exposure to radiation during the imaging procedure, which may slightly increase the risk of developing cancer [129].

#### 2.2.2. Clinical and Scientific Relevance of CTA for Ocular and Cardiovascular Diseases

CTA is a rapid and reliable method for obtaining precise and reproducible morphological information on blood vessels [130]. With an increasing number of image sections per study, it allows for visualization of complex vascular structures and smaller vessels [131]. Morphometric studies have been conducted using CTA to accurately determine vessel diameter of the OA [131]. The bilateral measurements of the OA were taken at its entrance into the optical canal, while the internal carotid artery was measured both caudally and cranially from the origin of the OA [131]. The diameter values found were consistent with the data from other studies that analyzed OA vessel diameter through microsurgical investigations [132,133]. No significant differences in OA vessel diameter were found between dominant and non-dominant sides [131]. Similarly, other authors utilizing CTA to determine OA vessel diameter reported no major variations related to gender or age [134]. Furthermore, CTA can be used to examine underlying anatomical and morphological variations in ocular and cardiovascular diseases. For instance, Rossin et al. discovered that patients with retinal ischemia of embolic origin exhibited a more proximal takeoff of the OA from the ICA compared to patients with stroke or transient ischemic attack (TIA) [135]. In patients with acute coronary syndrome (ACS), hemodynamic features of the OA were studied through computational fluid dynamics simulations to generate streamline charts for each 3D model of the OA [136,137]. The blood flow velocity in the OA was lower in patients with ACS and correlated with clinical parameters associated with ACS [137]. In patients with type 2 diabetes, with or without ACS, all disease groups displayed lower mass flow rates and blood flow velocities in the OA compared to control groups, and patients with ACS exhibited lower flow rates than patients with diabetes alone [136]. Additionally, patients with diabetes demonstrated a smaller OA vessel diameter compared to patients with ACS only [136]. The hemodynamic parameters collected in this study were correlated with several clinical indicators, suggesting a prognostic value of the OA hemodynamic characteristics in diabetes and ACS [136].

Although CTA has been used to analyze the vasculature in mice, such as coronary arteries [138], there have been no published CTA studies on retrobulbar vessels in laboratory animals. The optical resolution required for CTA in laboratory animals poses a challenge, and the high heart rate of small laboratory animals complicates vascular studies. Analysis of angiogenesis in a rabbit VX2 tumor revealed a minimum detectable vessel diameter of 0.68 ± 0.07 mm by three dimensional CTA, compared to a minimum detectable vessel diameter of 0.85 ± 0.12 mm using four-dimensional contrast-enhanced magnetic resonance angiography (MRA) [139].

In conclusion, CTA is a valuable tool for obtaining detailed morphological information on the OA, including vessel diameter measurements at various locations along its course. Representative CTA images showing the course of the OA are demonstrated in Figure 3. However, it is important to consider the potential adverse effects associated with intravenous contrast administration and the limitations in functional studies.

### 2.3. Magnetic Resonance Imaging

#### 2.3.1. Technical Aspects

Magnetic resonance imaging (MRI) is a non-invasive imaging modality that relies on a magnetic field to align the protons within water molecules in the body. By applying a radio wave, these protons are excited and emit energy signals that are captured by the MRI scanner. This technique produces high-resolution 3D images with superior soft tissue contrast compared to CT scans or X-rays [140]. One significant advantage of MRI is that it does not involve radiation exposure. However, there are certain drawbacks associated with this technique. Exposure to noise during image acquisition and claustrophobia can be a limiting factor for some patients. Additionally, the presence of electronic implants such as pacemakers can pose a challenge or even prohibit an MRI scan due to the strong magnetic fields. In rare cases, intravenous administration of Gadolinium-based contrast agents may lead to nephrogenic systemic fibrosis (NSF) in individuals with severe renal disease [141]. However, methods exist to avoid the use of contrast agents in MRA. For example, phase-contrast MRI is a viable method for assessing blood flow rates and has been extensively utilized in larger vessels, such as the aorta or intracerebral vessels [142,143,144,145].

#### 2.3.2. Clinical and Scientific Relevance of MRI for Ocular and Cardiovascular Diseases

MRI has also been introduced for the evaluation of blood flow rates in retrobulbar vessels [146]. Using phase-contrast MRI, detailed morphological classification of the OA into straight, curved, and tortuous types can be made, and the angle between the ICA and the OA can be determined [147]. MRI allows also for exploration of the proximal part of the CRA and the SOV [148,149]. Furthermore, MRI has the capability to depict complex anatomical variations in retrobulbar vessels [150].

Impaired retrobulbar blood flow has also been identified in patients with age-related macular degeneration. Ultra-high-field MRI has revealed significantly smaller OA vessel diameters, lower OA volumetric blood flow, higher RI, and higher OA flow velocities in these patients. Remarkably, these findings correlate with increasing disease severity [151].

Disturbances in retrobulbar blood flow have also been observed in patients with glaucoma in MRI studies. Parameters such as the pulsatility index (PI), RI, peak systolic flow (PSF), and minimum diastolic flow (MDF) velocities of the OA can be calculated based on the mean blood flow rate during the cardiac cycle. A tendency of reduced blood flow rates in patients with NTG compared to healthy individuals has been found, and the blood flow rate was reported to be even lower in the worse eye [146]. Other studies have reported similar results, with reduced mean and maximal flow in the OA and SOV in patients with open-angle glaucoma. Furthermore, a reduced variation of flow in the SOV has been detected in the glaucoma group, supporting the hypothesis of impaired venous outflow in these patients [152]. However, changes in blood flow rates of the OA after significant IOP reduction of in patients with ocular hypertension could not be detected [153].

In patients with high-grade cervical carotid stenosis (70–99%) or occlusion, reversed OA flow has been detected in MRI studies. The combination of reversed OA flow, history of stroke, and intracranial stenosis has been shown to be a significant risk factor for poor functional outcomes [154]. Other studies have also found an increased incidence of intracranial vessel stenosis and reversed OA flow in patients with acute stroke and severe unilateral cervical carotid stenosis. Moreover, reversed OA flow has been associated with a 10–20% improvement in stroke outcome compared to patients with forward OA flow and the same degree of intracranial stenosis [155]. Analysis of OA flow direction may also help to predict intracranial hemodynamic impairment, since the ratio of cerebral blood flow at rest to normal control and cerebral vascular reserve is reduced in patients with non-native OA flow [156]. Diameter reduction and decreased blood flow velocities of the OA have shown strong associations with cerebral small vessel disease and the presence of cerebral microbleeds, enlarged perivascular spaces, lacunar infarcts and white matter hyperintensities. These findings suggest that observing OA characteristics may enable the evaluation of the vascular status and diseases of the brain [157].

The implementation of MRI angiography in laboratory animals has frequently been performed to analyze the brain vasculature [158,159]. Recently, the application of MRI for evaluating blood flow in the brain and OA was reported in a mouse model of cerebral ischemia, and an anatomical origin of the OA from the pterygopalatine artery was detected in this context. Moreover, comparison of two different methods of middle cerebral artery occlusion was possible [160]. MRI is a non-invasive but expensive method that requires significant technical effort to evaluate retrobulbar blood flow. Its use in laboratory animals is challenging due to small vessel size, and results of vascular studies may be affected by general anesthesia.

### 2.4. Evaluation of Retrobulbar Vessels Using Myography

Although non-invasive in vivo measurement techniques of retrobulbar hemodynamics are valuable, they have certain technical limitations that can impede the interpretation of experimental findings [27,161]. For example, the effects of anesthesia, fluctuations in systemic blood pressure, and regulatory mechanisms of the retinal and choroidal blood vessels may confound the interpretation of in vivo measurements aimed at identifying local signaling mechanisms in retrobulbar blood vessels. Therefore, from a scientific point of view, studies specifically designed for measurement of vascular reactivity in isolated retrobulbar blood vessels are crucial to understanding the local mechanisms of blood flow regulation under both healthy and pathological conditions. Isolation and measurement of retrobulbar vessels, such as ciliary arteries and the OA, have been reported previously for larger species, including humans, pigs, cattle, and rabbits [162,163,164,165]. The relatively large luminal diameter of these vessels, usually above 80 µm, allows for measurement by wire or pressure myography. These techniques allow for the investigation of vascular responses in isolated retrobulbar blood vessels at standardized conditions providing valuable insights into the physiological and pathophysiological mechanisms underlying blood flow regulation in the retrobulbar region. By conducting ex vivo experiments under controlled conditions, local factors influencing retrobulbar hemodynamics without the confounding influence of systemic factors can be investigated. The use of wire or pressure myography enhances our ability to observe and analyze vascular dynamics and responses in real time, offering a detailed understanding of the retrobulbar vascular physiology. In recent years, the mouse has emerged as a widely used animal model for cardiovascular research, owing to the advent of gene manipulation in the murine genome [166]. However, accessing retrobulbar blood vessels in mice poses challenges due to their small size, extensive branching, and their location within the intricate orbital tissue structures, including bones. To address this, we have developed an isolation technique that enables ex vivo studies of the mouse OA under near-physiological conditions using transmitted light microscopy.

#### 2.4.1. Technical Aspects

The first step in this procedure is the isolation of the OA, as described previously [167]. An easy way to find the OA is to expose it at its entrance into the bulbar wall by removing the extraocular muscles and the connective tissue surrounding the optic nerve. In pigmented mice, the artery is easily detectable by its pigment deposits along the vascular wall (Figure 4). The artery can then be tracked to its proximal part of larger diameter, which is not attached to the optic nerve, and is therefore easier to isolate.

For functional experiments, commercially available or self-made perfusion chambers can be used. We used chambers obtained from Jim’s Instruments Manufacturing Inc., Iowa City, IA, USA. The lumen of the isolated vessel segment is cannulated by the tips of a micropipette and tied with 10.0 nylon suture. The perfusion chamber is then placed under a microscope and connected to the circulatory system for the experiment. The vessel segment is then visualized, and its responses are recorded by using video editing software. Diameter changes in the vessel can be measured by a screen ruler or by automated measuring systems. For a better understanding of the experimental setup see the detailed sketch provided in Figure 5.

By using this method, small changes in vascular diameter in response to vasoactive substances or pressure stimuli can be detected and recorded by video microscopy. We previously studied the mechanisms of vascular smooth muscle and endothelial function of the mouse OA and pig ciliary arteries by using this method and compared the results with measurements in retinal vascular preparations. Interestingly, we found a variety of functional differences between the retinal arterioles and retrobulbar vessels within one species. For example, in the mouse OA, responses to α_1_-adrenergic stimuli were predominantly mediated by the α_1A_-adrenoceptor, whereas in retinal arterioles the α_1B_-adrenoceptor mainly mediated these responses [168,169]. Endothelium-dependent vasodilation was mediated in part by endothelial nitric oxide synthase (eNOS) and by endothelium-derived hyperpolarizing factors (EDHFs) in the mouse OA [170,171,172]. In contrast, endothelial vasodilatory responses in mouse retinal arterioles were predominantly mediated by eNOS and partially preserved by neuronal nitric oxide synthase (nNOS) and cyclooxygenase 2 (COX2) during eNOS deficiency [173,174]. Likewise, retinal arterioles from pigs with acute respiratory distress syndrome (ARDS) developed remarkable endothelial dysfunction, whereas endothelium-dependent ciliary artery responses were preserved [163]. These findings demonstrate profound differences in vascular signaling mechanisms between retinal and retrobulbar blood vessels highlighting the usefulness of myography to determine local mechanisms of vascular function.

#### 2.4.2. Advantages and Limitations of Myography

Myography allows for direct measurement of vascular function under standardized experimental conditions without the need for fluorescent dye. In our own studies, pharmacological agents were applied extraluminally, but intraluminal perfusion using a servo control pump is also a viable option. With this technique, vessels from small laboratory animals, such as mice and rats, can also be measured, which offers the advantage of accessing gene-targeting technology. However, some critical steps in this technique require special attention. First, it is critical to mount the vessel within a timeframe of several hours. We observed impaired responses when retrobulbar vessels were studied after more than six hours after the research subject’s death. This makes it nearly impossible to conduct such experiments in deceased humans. Although studies in human vessels can be conducted in tissue from enucleated eyes, these eyes typically have other pathologies, which may affect the experimental results. Second, for experiments using pressure myography, it is crucial to either select a vascular segment without side branches or to ligate all branches to avoid leakage, which may induce flow through the vessel leading to shear stress at the luminal wall. Shear stress induces a release of endothelial vasoactive factors such as nitric oxide, which can affect the experimental data.

### 2.5. Clinical and Scientific Relevance of Measuring Retrobulbar Blood Flow/Vascular Reactivity and Future Directions

Measurement of retrobulbar blood flow is important for understanding the role of hemodynamic changes in retrobulbar vessels in the pathogenesis of several ocular diseases as described above. Vascular dysfunction is discussed to play a central role in several ocular diseases, such as glaucoma [175]. Table 1 provides an overview of the described methods.

CDI is a non-invasive and fast method of evaluating retrobulbar blood flow velocities. CDI studies have been used in this context to analyze the pathogenesis of several ocular diseases. Reduced retrobulbar blood flow velocities have been observed in patients with NTG or POAG [1,176]. Interestingly, a prospective study conducted in patients with glaucoma demonstrated a notable association between altered blood flow in retrobulbar vessels and changes in optic nerve blood flow. This finding suggests a potential link between retrobulbar hemodynamics and optic nerve perfusion in individuals with glaucoma [9]. Cardiovascular disease is often associated with hypoperfusion of ocular structures, leading to severe vision loss, as seen in non-arteritic ischemic optic neuropathy. Reduced blood flow velocities in the posterior ciliary arteries and CRA have been detected in patients with this condition [4,5]. Central retinal artery occlusion (CRAO), primarily caused by embolic mechanisms from the ICA, also results in severe vision loss. Alterations in retrobulbar blood flow have been observed in individuals affected by CRAO [177]. Furthermore, in patients with ocular diseases, disruptions in the autoregulatory mechanisms of retrobulbar vessels may occur [178,179]. In addition to the analysis of pathogenetic aspects, CDI has also been used to evaluate the impacts of different ocular therapies [180,181,182]. For example, the effects of antiglaucomatous drugs could be analyzed using this method [180]. As described above, there are studies that found changes in retrobulbar hemodynamics after application of antiglaucomatous drugs, such as travoprost [70], and others that found no differences of retrobulbar blood flow velocities after antiglaucomatous therapy [69]. Schmetterer et al. observed reduced choroidal and optic disc blood flow after application of timolol or clonidine, while retrobulbar blood flow velocities remained stable. Autoregulatory mechanisms might explain these findings. But these findings indicate that combination of retrobulbar blood flow measurement with measurement of choroidal and optic disc blood flow is necessary for understanding the complex vascular system of ocular structures [180]. CDI was also used to analyze the effects of panretinal photocoagulation on retrobulbar hemodynamics in diabetic patients, and higher blood flow velocities after treatment were detected [181]. Moreover, in neovascular age-related macular degeneration intravitreal injections of ranibizumab or aflibercept lead to reduced blood flow velocities in retrobulbar vessels [182,183].

In addition to ocular diseases, cardiovascular conditions such as heart failure or carotid artery stenosis have been shown to impact retrobulbar hemodynamics. Heart failure, characterized by sodium-water retention, congestion, decreased systemic blood pressure, and increased orbital vasoconstriction, is associated with reduced retrobulbar blood flow velocities detected by CDI [101]. Furthermore, changes in blood flow within the OA have been observed in patients with severe ICA occlusion and acute stroke [155].

These findings suggest that alterations in retrobulbar blood flow may not only be limited to ocular diseases but also reflect significant systemic cardiovascular conditions. Understanding the regulation of retrobulbar blood flow and its disruptions in ocular diseases is crucial for comprehending the pathogenesis of various ocular conditions and developing potential therapeutic strategies. Additionally, analyzing blood flow in retrobulbar vessels may aid in predicting the risk of cardiovascular events, such as stroke or acute coronary syndromes.

All these findings indicate that CDI is a feasible and non-invasive method of evaluating retrobulbar blood flow velocities with the possibility of performing long-term studies. But this method lacks the possibility of acquiring volumetric data and information about vessel diameters. In addition, varying the angle of incidence may affect reproducibility and exact knowledge about retrobulbar anatomy is important.

Other methods, such as CTA and MRI, are also useful for analyzing retrobulbar blood flow. By conducting CTA and MRI scans, more detailed information about retrobulbar anatomy and the course of these vessels is achieved compared to CDI. Moreover, in addition to measurement of blood flow rates and velocity, determination of vessel diameter is possible. In comparison to CDI, higher technical effort is necessary using these methods, and CTA in particular is more invasive due to the need for intravenous application of a contrast dye. For studies in animal models with the possibility of genetic modification limited optical resolution may hamper these studies. In three-dimensional CTA, a minimum detectable vessel diameter of 0.68 ± 0.07 mm and in four-dimensional contrast-enhanced MRA of 0.85 ± 0.12 mm was found [139]. In this study, CTA revealed better spatial resolution and MRA showed higher temporal resolution [139]. The mouse OA has a vessel diameter of about 50 to 150 µm and detection of diameter changes is not possible using this method [171]. Moreover, to perform experiments in laboratory animals using CTA or MRI, general anesthesia is required, which may influence the results of the study. Therefore, myography may represent a feasible method for performing vascular studies in these small animals, where diameter changes of 1–3% are detectable in ex vivo studies and no general anesthesia is required [39]. However, using this method, no longitudinal studies can be performed and experienced investigators are mandatory.

Table 2 summarizes the advantages and limitations of the different presented methods.

## 3. Conclusions

In conclusion, there are several methods to evaluate retrobulbar blood flow in vivo. CDI represents a fast and non-invasive method to measure retrobulbar blood flow velocities. But due to the lack of diameter determination, no volumetric blood flow data can be collected. Moreover, several sources of error, such as the angle of incidence, are described. CTA and MRA represent a more exact method to evaluate retrobulbar blood flow with the possibility of measuring vessel diameters and volumetric blood flow. Limitations of these methods are the high levels of technical effort required and the intravenous application of a contrast agent that has been related to several health risks. Moreover, studies in laboratory animals have barely been described so far. For functional studies of vascular reactivity, myography of the OA via transmitted light microscopy can be performed ex vivo in laboratory animals, including genetically modified mice. With this method, exact determination of vascular diameter is possible and ex vivo studies with concentration–response curves under near-physiological conditions can be performed. However, isolation of retrobulbar vessels is challenging and there are several critical steps to be considered with this technique. Taken together, this is the first review article summarizing available methods to measure retrobulbar blood flow and vascular reactivity for clinical as well as for experimental investigators. All of the presented methods have their advantages and limitations, and the preferable method should be chosen depending on the design and objective of the study.

## Figures and Tables

**Figure 1 diagnostics-13-03514-f001:**
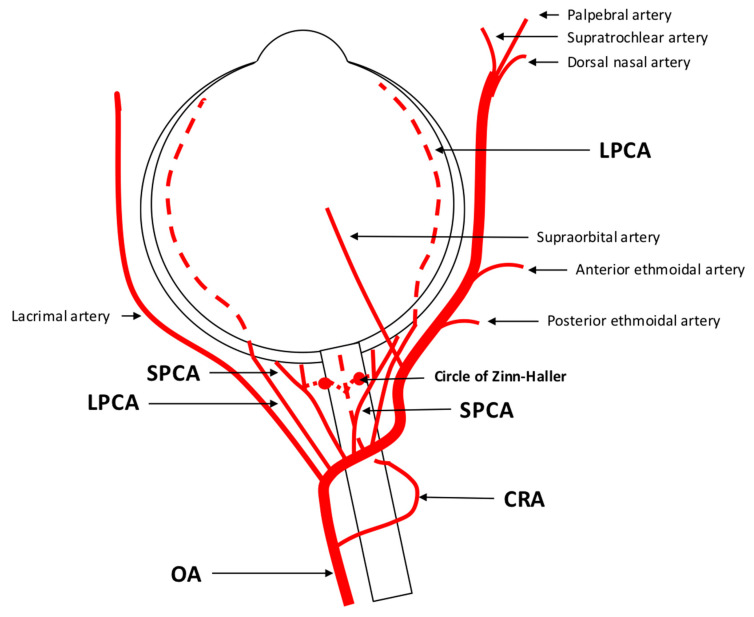
Anatomy of retrobulbar blood supply in a left eye. The ophthalmic artery (OA) serves as the initial branch of the intracranial internal carotid artery (ICA) and traverses the optic nerve, extending from an inferotemporal to a nasal direction. The first branch arising from the OA is the central retinal artery (CRA) responsible for supplying the inner retinal layers. The choroid receives its blood supply from the long and short posterior ciliary arteries (LPCA/SPCA). These arteries have the potential to form anastomoses with the CRA through the circle of Zinn-Haller. In addition to these branches, the OA gives rise to other vessels such as the lacrimal artery, supraorbital artery, posterior and anterior ethmoidal arteries, palpebral artery, dorsal nasal artery, and supratrochlear artery.

**Figure 2 diagnostics-13-03514-f002:**
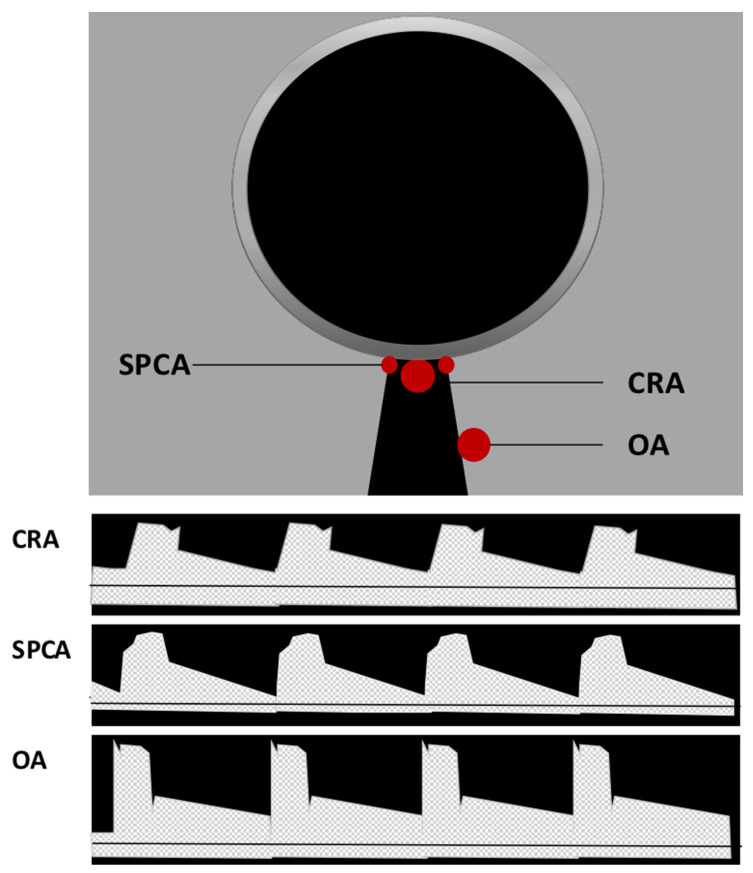
Location and Doppler waveform of retrobulbar vessels in Color Doppler imaging (CDI). Central retinal artery (CRA) and central retinal vein (CRV) can be identified in the middle of the optic nerve. Due to their proximity, their signal cannot be separated, and a double waveform is produced. Located on the respective sides of the optic nerve, nasal and temporal short posterior ciliary arteries (SPCA) can be detected generating a uniform arterial pulse representing a small bundle of vessels. Deeper in the retrobulbar tissue on the temporal side of the optic nerve, the ophthalmic artery (OA) flow parameters can be measured.

**Figure 3 diagnostics-13-03514-f003:**
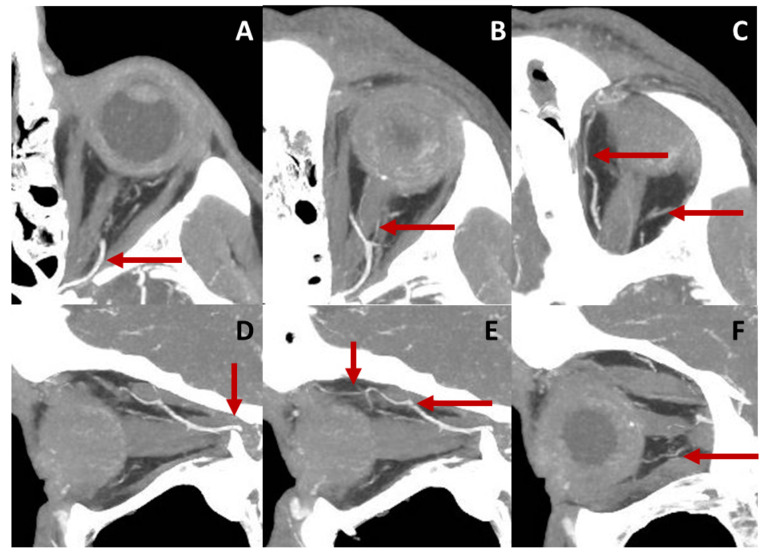
Course of the OA in a left eye displayed by computed tomography angiography (CTA). Axial sections of the retrobulbar anatomy are represented in the upper row. Image (**A**) shows the ophthalmic artery (OA) entering the orbit and crossing the optic nerve from its temporal to its nasal side (**B**). Central retinal artery (CRA), as a branch of the ophthalmic artery (OA), can be detected in (**B**). The OA follows a nasal course and gives rise to other branches, such as the lacrimal and supraorbital arteries (**C**). Sagittal sections of the orbit are presented in the lower row. The OA enters the orbit traversing the optic canal (**D**). It crosses over the optic nerve (**D**,**E**) and releases branches, such as ethmoidal arteries, the supraorbital artery (**E**) or posterior ciliary arteries (**F**).

**Figure 4 diagnostics-13-03514-f004:**
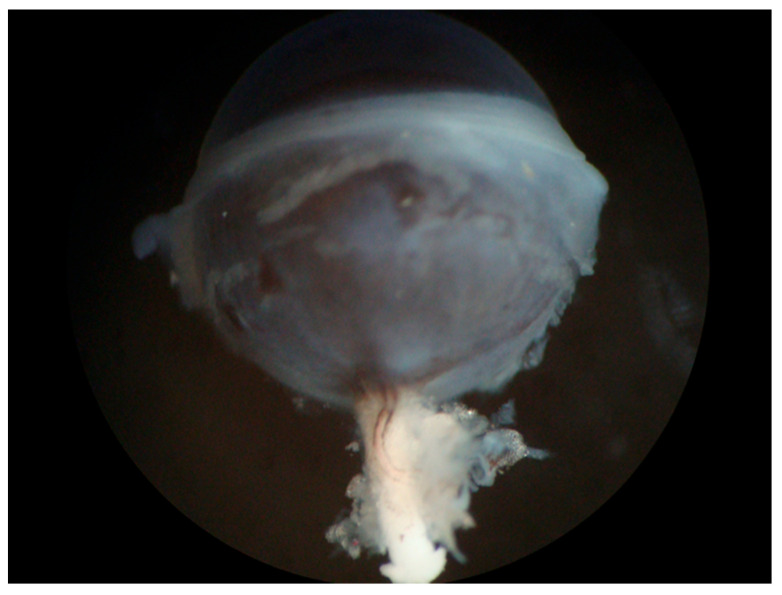
Eye of a pigmented mouse (C57BL/6J) with the optic nerve and visible ophthalmic artery (OA). The OA can be easily recognized on the white optic nerve by its dark pigment deposits along the vascular wall.

**Figure 5 diagnostics-13-03514-f005:**
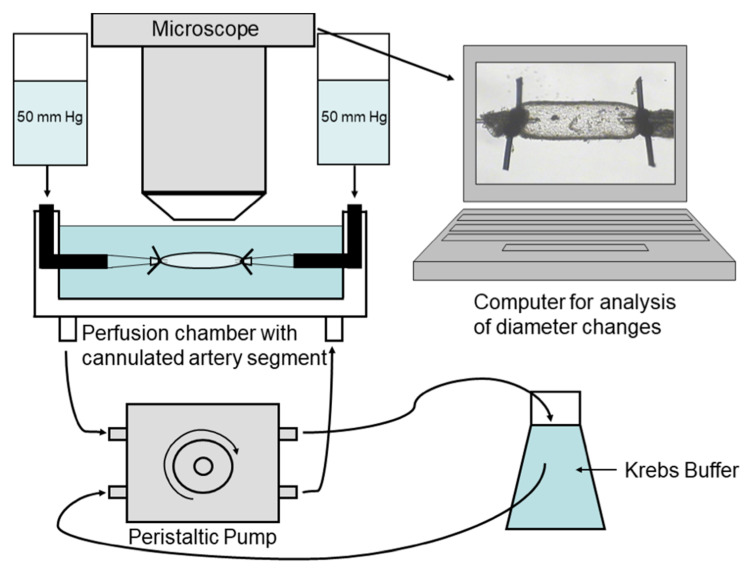
Experimental setting. A segment of the ophthalmic artery is placed in a perfusion chamber, cannulated from both ends with glass micropipettes, tied with 10.0 nylon suture, and placed under a microscope to record vascular responses.

**Table 1 diagnostics-13-03514-t001:** Study condition, principle of measurement, and main content of the presented methods.

Methods	Study Condition	Principle of Measurement	Main Content
CDI	In vivo	Combination of B-scan ultrasonography with Doppler technology	Anatomical information Blood flow velocity Visualization of blood flow direction
CTA	In vivo	X-rays and advanced computer processing to generate detailed cross-sectional images	Morphological information Determination of vessel diameter
MRI	In vivo	Magnetic field to align the protons within water molecules in the body	Morphological information Determination of vessel diameter Volumetric blood flow Blood flow velocity
Myography	Ex vivo	Ophthalmic artery isolation and measurement of vascular reactivity by transmitted light microscopy	Vessel diameter Vascular reactivity

**Table 2 diagnostics-13-03514-t002:** Advantages and limitations of the presented methods.

Methods	Advantages	Limitations
CDI	Commercially available	No volumetric blood flow data
	Non-invasive and rapid method	No information about vessel diameter
	Studies in laboratory animals available	Angle of incidence as common source of error
		Training mandatory to ameliorate reproducibility
CTA	Rapid and reliable method	Potential adverse effects associated with intravenous contrast agents
	Reproducible data on blood vessel morphology and course	Limitation in functional studies
		Radiation exposure
		Limited use in laboratory animals due to small vessel size and affection by general anesthesia
MRI	Non-invasive	Expensive
	High-resolution 3D images with superior soft tissue contrast	Potential adverse effects associated with intravenous contrast agents
	Methods without use of contrast agents available	Exposure to noise, claustrophobia and presence of electronic implants as limiting factors for patients
	No radiation exposure	Limited use in laboratory animals due to small vessel size and affection by general anesthesia
Myography	Investigation of local mechanisms of vascular reactivity in small laboratory animals	Difficult isolation and preparation of retrobulbar vessels
	No affection by general anesthesia in laboratory animals	No longitudinal studies available
	Studies in genetically modified animals available	

## Data Availability

Since no new data were created or analyzed in this study, data sharing is not applicable to this article.

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
