# Peer review of "Measurement of Retrobulbar Blood Flow and Vascular Reactivity—Relevance for Ocular and Cardiovascular Diseases"

_diagnostics, 2023, doi:10.3390/diagnostics13233514_

Round 1
Reviewer 1 Report
Comments and Suggestions for Authors
Provide the obtained results and metrics at end of the abstract.
Author has to justify the dataset taken and its necessity in the current research.
In section 2.4. Evaluation of Retrobulbar Vessels Using Myography- How the evaluation has carried? Are you using any tools?
What is the source of all the sample images?
The motivation must be given clearly in the Introduction section.
The authors should clearly justify the need for their approach in the presence of many effective approaches and should clearly state the pros and cons of their suggested technique.
The Introduction should make a compelling case for why the study is helpful along with a clear statement of its novelty or originality by providing relevant information and answering basic questions such as: What is already known in the open literature? What is missing (i.e., research gaps)? What needs to be done, why, and how? Clear statements of the novelty of the work should also appear briefly in the Abstract and Conclusions sections.
Author Response
- Provide the obtained results and metrics at end of the abstract.
Authors’ response to 1.) Thank you for this comment. We added results and metrics at the end of the abstract. The corrected abstract reads as follows:
“…Various methods, such as color Doppler imaging, computed tomography angiography or magnetic resonance imaging, have been employed to assess retrobulbar blood flow velocities in vivo. Color Doppler imaging represents a fast and non-invasive method to measure retrobulbar blood flow velocities in vivo. While no information about vessel diameter can be assessed performing this method, computed tomography angiography and magnetic resonance imaging provide information about vessel diameter and detailed information about the anatomical course. Additionally, ex vivo studies, such as myography, utilizing genetically modified animal models may provide useful, necessitating high optical resolution for functional vascular investigations in these small vessels.”
- Author has to justify the dataset taken and its necessity in the current research.
Authors’ response to 2.) Thank you for this helpful advice. We now mentioned how the dataset was taken in the manuscript: “The literature was identified via search on PubMed. The following keywords were used: (“retrobulbar blood flow” OR “ophthalmic artery” OR “central retinal artery” OR “posterior ciliary artery”) AND (“measurement” OR “color Doppler imaging” OR “computed tomography angiography” OR “magnetic resonance imaging” OR “myography” OR “transmitted light microscopy”). The research was performed from 11 March 2023 to 30 June 2023 with the following inclusion criteria: all studies, written in English, and published after 1990. The reference list of the selected articles was reviewed for further identification of relevant studies.”
- In section 2.4. Evaluation of Retrobulbar Vessels Using Myography- How the evaluation has carried? Are you using any tools?
Authors’ response to 3.) Thank you for this advice. The vessel segment is visualized, and its responses are recorded by using a video editing software. We used first Khoros from Khoral Research Inc. and later PowerDirector from CyberLink. However, many other softwares are available depending on the operating system used, which can be used for this purpose. Diameter changes of the vessel can be measured by a screen ruler or by automated measuring systems. We included these statements in the manuscript (lines 338-339).
- What is the source of all the sample images?
Authors’ response to 4.) Thank you for this comment. The presented samples in section 2.4. Evaluation of Retrobulbar Vessels Using Myography are unpublished and were taken in our own laboratory.
- The motivation must be given clearly in the Introduction section. The authors should clearly justify the need for their approach in the presence of many effective approaches and should clearly state the pros and cons of their suggested technique.
Authors’ response to 5.) Thank you for this helpful comment. We now offered a more detailed description of the motivation and the need for our approach in the introduction section:
“This study should help clinical as well as experimental investigators to identify their gaps in knowledge and define suitable methods designing their research projects in the field of retrobulbar blood flow.”
Moreover, we added Table 2, where we present the advantages and limitations of the presented techniques.
- The Introduction should make a compelling case for why the study is helpful along with a clear statement of its novelty or originality by providing relevant information and answering basic questions such as: What is already known in the open literature? What is missing (i.e., research gaps)? What needs to be done, why, and how? Clear statements of the novelty of the work should also appear briefly in the Abstract and Conclusions sections.
Authors’ response to 6.) Thank you for this comment. We added more information of the novelty and originality of the study in the Introduction section. The sentence reads as follows:
“To the best of our knowledge, this is the first comprehensive review summarizing contemporary state of the art regarding retrobulbar blood flow measurement under physiologic and pathologic conditions.”
We mentioned the novelty and originality of the review article also in the Abstract and Conclusion section. Please refer to line 26-27, line 458-459.
Reviewer 2 Report
Comments and Suggestions for Authors
This is a good review article on the clinical application of blood flow measurements and their associated techniques. In particular, the findings from CDI, CT angiography, and myography using isolated retrobulbar vessels are quite suggestive.
Comments on the Quality of English LanguageThere are very few typographical errors present.
Author Response
- This is a good review article on the clinical application of blood flow measurements and their associated techniques. In particular, the findings from CDI, CT angiography, and myography using isolated retrobulbar vessels are quite suggestive.
Authors’ response to 1.) Thank you. We appreciate this comment.
- There are very few typographical errors present.
Authors’ response to 2.) Thank you for this comment. We corrected topographical errors.
Reviewer 3 Report
Comments and Suggestions for Authors
On the whole, I think this review is well written.But I still have some suggestions.
First, there are some minor grammatical mistakes in the article. Or places where the expression is not smooth.
I strongly suggest that the author consider using tables to summarize related contents to give readers a better reading experience.
Comments on the Quality of English Language
Minor editing of English language required
Author Response
- On the whole, I think this review is well written. But I still have some suggestions. First, there are some minor grammatical mistakes in the article. Or places where the expression is not smooth.
Authors’ response to 1.) Thank you for this comment. We corrected grammatical mistakes and revised wording at some points.
- I strongly suggest that the author consider using tables to summarize related contents to give readers a better reading experience.
Authors’ response to 2.) Thank you for this comment. We added Table 1 and Table 2 to summarize the different presented methods and their advantages and limitations.
Table 1. Study condition, principle of measurement and main content of the presented methods.
Methods |
Study condition |
Principle of measurement |
Main content |
CDI |
In vivo |
Combination of B-scan ultrasonography with Doppler technology |
Anatomical information Blood flow velocity Visualization of blood flow direction |
CTA |
In vivo |
X-rays and advanced computer processing to generate detailed cross-sectional images |
Morphological information Determination of vessel diameter |
MRI |
In vivo |
Magnetic field to align the protons within water molecules in the body |
Morphological information Determination of vessel diameter Volumetric blood flow Blood flow velocity |
Myography |
Ex vivo |
Ophthalmic artery isolation and measurement of vascular reactivity by transmitted light microscopy |
Vessel diameter Vascular reactivity |
|
|
|
Table 2. Advantages and Limitations of the presented methods.
Methods |
Advantages |
Limitations |
CDI |
Commercially available |
No volumetric blood flow data |
Non-invasive and rapid method |
No information about vessel diameter |
|
Studies in laboratory animals available |
Angle of incidence as common source of error |
|
|
Training mandatory to ameliorate reproducibility |
|
CTA |
Rapid and reliable method |
Potential adverse effects associated with intravenous contrast agents |
Reproducible data on blood vessel morphology and course |
Limitation in functional studies |
|
|
Radiation exposure |
|
Limited use in laboratory animals due to small vessel size and affection by general anesthesia |
||
MRI |
Non-invasive |
Expensive |
High-resolution 3D images with superior soft tissue contrast |
Potential adverse effects associated with intravenous contrast agents |
|
Methods without use of contrast agents available |
Exposure to noise, claustrophobia and presence of electronic implants as limiting factors for patients |
|
No radiation exposure |
Limited use in laboratory animals due to small vessel size and affection by general anesthesia |
|
Myography |
Investigation of local mechanisms of vascular reactivity in small laboratory animals |
Difficult isolation and preparation of retrobulbar vessels |
No affection by general anesthesia in laboratory animals |
No longitudinal studies available |
|
|
Studies in genetically modified animals available |
|
Reviewer 4 Report
Comments and Suggestions for Authors
I extend my gratitude to the authors of this article for providing a comprehensive overview in accessible language. The article is notably engaging as it synthesizes the pivotal methodologies for the examination of vascular parameters. Except for myography and the part about ex vivo, which seems incongruent with the overall narrative, the article adeptly elucidates the methodologies, offering insights into the parameters that can be derived, the merits and demerits of each approach, and the attendant limitations necessitating consideration during their utilization. The article also impressively delves into clinical case studies pertaining to vascular issues and previous research employing these methodologies.
Nonetheless, the article would benefit from a grammatical review, as it occasionally contains clerical errors and instances of word repetition.
I would like to propose, however, that the inclusion of myography (the part about ex vivo measurements) be reevaluated. It stands apart as an entirely distinct methodology, typically applied to post-mortem research subjects, primarily animals in this context. Consequently, its incorporation seems incongruous with the article's focus, which centers on methodologies applied to living subjects for the purpose of timely diagnosis and the detection of vascular issues.
Author Response
- I extend my gratitude to the authors of this article for providing a comprehensive overview in accessible language. The article is notably engaging as it synthesizes the pivotal methodologies for the examination of vascular parameters. Except for myography and the part about ex vivo, which seems incongruent with the overall narrative, the article adeptly elucidates the methodologies, offering insights into the parameters that can be derived, the merits and demerits of each approach, and the attendant limitations necessitating consideration during their utilization. The article also impressively delves into clinical case studies pertaining to vascular issues and previous research employing these methodologies.
Authors’ response to 1.) Thank you for this comment.
- Nonetheless, the article would benefit from a grammatical review, as it occasionally contains clerical errors and instances of word repetition.
Authors’ response to 2.) Thank you for this comment. We performed grammatical revision and corrected errors as wells as word repetitions.
- I would like to propose, however, that the inclusion of myography (the part about ex vivo measurements) be reevaluated. It stands apart as an entirely distinct methodology, typically applied to post-mortem research subjects, primarily animals in this context. Consequently, its incorporation seems incongruous with the article's focus, which centers on methodologies applied to living subjects for the purpose of timely diagnosis and the detection of vascular issues.
Authors’ response to 3.) Thank you for this helpful comment. It is well taken! We understand that the presented ex vivo method in the section 2.4. Evaluation of Retrobulbar Vessels Using myography stands a bit apart from the presented in vivo methods. The intention of this review article was, however, to give a total overview on available methods to measure retrobulbar blood flow and vascular reactivity to clinical as well as to experimental investigators. To fully understand the complex regulation of vascular reactivity in the retrobulbar vasculature, ex vivo studies in laboratory animals with the possibility of genetic modification may be helpful to analyze molecular mechanisms in this field. We shortened the section a bit to make it congruent with the other sections.